# Responses of Terrestrial Evapotranspiration to Extreme Drought: A Review

**Qiu-Lan He, Jun-Lan Xiao and Wei-Yu Shi \***

Chongqing Jinfo Mountain Karst Ecosystem National Observation and Research Station, School of Geographical Sciences, Southwest University, Chongqing 400715, China
*   Correspondence: shiweiyu@swu.edu.cn; Tel.: +86 23-68253912

**Abstract:** Terrestrial evapotranspiration (ET) is crucial to the exchange of global carbon, water, and energy cycles and links the hydrological and ecological processes. The frequency and intensity of extreme droughts are expected to increase due to ongoing climate change, strongly impacting terrestrial ET with implications for ecosystems, societies, and climate systems. However, the response of terrestrial ET to extreme drought and the underlying mechanism of terrestrial ET change during droughts are still unclear. Here, we review previous studies on terrestrial ET's responses to extreme drought and investigate the control factors of ET change in response to extreme drought under different situations. The response of terrestrial ET to extreme drought is affected by various factors including the duration and intensity of the drought, the original climate conditions, as well as the plant species. Terrestrial ET change during droughts is controlled by complex biological and physical processes that can be divided into four parts including supply, energy, demand, and vegetation activities. The response of terrestrial ET to elevate $CO_2$ may offset the effects of drought because $CO_2$ fertilization tends to increase water use efficiency through stomatal regulation. We found that large uncertainties remain in the terrestrial ET response to drought due to the discrepancies among different ET products and simulations. This work highlights the requirement for accurate estimates of ET changes in ET products and models. This review provides a systematic investigation of the terrestrial ET response to extreme drought and the underlying mechanism of terrestrial ET changes during droughts and will significantly improve the development of water management strategies under climate change.

**Keywords:** terrestrial ET; drought; elevate $CO_2$; climate change; water use

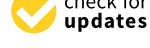

## 1. Introduction

Climate change is likely to intensify the water cycle and alter evapotranspiration (ET), potentially increasing the intensity and frequency of extreme droughts and floods, with implications for ecosystems, societies, and regional and global climate systems [1]. Terrestrial ET is the process of transporting water from the land surface to the atmosphere, which consists of abiotic evaporation from bare soil and open water bodies, etc., and biotic transpiration through leaf stomata [2,3]. ET is a critical process in the climate system and a key variable that links the water, energy, and carbon cycles, which recycle over 60% of land participation into the atmosphere [4,5]. It can affect precipitation, soil moisture, runoff, groundwater, and thus the water source availability [6,7]. Moreover, ET also has an impact on climate processes by influencing the partitioning of the sensible and latent heat fluxes, particularly on the surface temperature, and ultimately modifies the regional and global climate system [5]. Extreme drought is the most critical natural hazard, with increasing water resource scarcity and carbon cycle anomalies [8,9]. Higher temperatures and evaporation demand increase ET during droughts, accelerating surface water and groundwater depletion, and ultimately induce widespread plant mortality and carbon loss [10–12]. Indeed, changes in ET caused by extreme drought will have a

significant impact on the regional water resource proportion and earth's climate system due to the land–atmosphere interaction [13–15]. However, despite numerous studies on ET's response to extreme drought, the magnitude and sign of ET changes during drought remain unknow. Hence, because of the feedback of the drought-related changes in ET, it is urgent to investigate the underlying mechanism of ET changes during droughts.

Drought events are recognized as the most widespread and devastating natural hazard, which cause large-scale crop failures, an increase in tree mortality, and economic losses [8,16,17]. Numerous studies suggest that the intensity and frequency of extreme droughts are expected to increase under climate change [18,19]. However, terrestrial ET variations during droughts still remain uncertain and debatable due to the inconsistent response between the limiting water supply and the increasing evaporation demand. On the one hand, extreme droughts are assumed to decline ET by reducing the available water sources, which results in a decrease in both soil evaporation and plant transpiration [20]. On the other hand, extreme droughts may also increase ET by enhancing the transpiration rate through increasing the atmospheric evaporative demand [21], since transpiration is the major contributor to terrestrial ET [22,23]. In addition, the response of vegetation physiological activities is likely to increase the uncertainties of ET changes because plant stomatal dynamics and root growth would prevent water loss and thus limit plant transpiration [24]. What's more, the controlling factors of ET changes during drought are highly uncertain, particularly in transitional climate regions [20]. Previous studies have reported both increases and decreases in ET induced by drought events [25–27]. However, these studies have generally focused on specific drought events, regions, and species ET responses, without a global perspective on the response of terrestrial ET to extreme drought. In addition, large uncertainties existing among the different ET products and quantitative methods make it unclear whether these products and models can accurately capture the sign of ET changes during droughts. Therefore, there is a critical need to thoroughly understand the process of ET and the mechanisms of ET changes during droughts.

Since the population growth and industrial expansion, the scarcity of water resources has become extremely urgent [28]. Investigating the response of ET to extreme drought and the underlying mechanisms are essential to improve our understanding of the earth system and to manage water resources under climate change. In this review, we summarize the current understanding of terrestrial ET's response to extreme droughts and highlight the research gap in determining the controlling factors of terrestrial ET during droughts. In Section 2, we summarize the current methods to estimate or measure ET and discuss the uncertainties that remain in these methods. Section 3 discusses the process of terrestrial ET's response to extreme drought and the potential influencing factors. Section 4 presents the controlling factors for terrestrial ET changes during droughts, and the main conclusions are provided in Section 5.

## 2. Methods to Estimate and Measure Evapotranspiration

Multiple ET acquisition methods and ET products are used in studies of ET's response to extreme droughts at various spatial–temporal scales, which resulted in significant discrepancies in the findings. Understanding of the uncertainties of existing ET products and ET acquisition methods is a prerequisite for ET-related research. Therefore, in this section, we review the drought response-related ET acquisition approaches and ET products below.

### 2.1. Estimation and Measurement of Evapotranspiration

Numerous in situ techniques exist to measure the actual ET from point to region scales, including weighing lysimeters, eddy covariance, scintillometers, the Bowen ratio, sap flow, pan measurements, atmospheric water balances, and surface water balance techniques. Weighing lysimeters are the standard approach to measure ET without any assumptions and are often used to compare with other techniques [20]. Due to their costs and limited areal extent, however, there are very few long-term measurement records worldwide [29]. Eddy covariance provides a method for measuring ET over a diurnal cycle [30,31] and has

been deployed as part of the FLUXNET network which covers over 500 sites worldwide [32] (https://fluxnet.org/sites/site-summary/, accessed on 22 November 2022). However, one also should note the issue with the eddy covariance system's energy balance closing errors caused by the spatial heterogeneity of atmospheric and surface conditions [33]. Nonetheless, the FLUXNET dataset provides valuable observations of terrestrial hydrological variables and can be useful for identifying the ET variability under extreme droughts on the point and regional scales. For example, several studies have evaluated ET dynamics using long-term FLUXNET datasets [34,35]. However, their applicability may be limited for large areas as it is not feasible to achieve dense global coverage with such point instruments. Atmospheric water balances and surface water balances provide a robust estimate of ET at the basin to global scales with a low temporal (yearly) resolution based on precipitation and streamflow observations [29]. These approaches are simple and theoretically reliable, but one should also note the uncertainties introduced by the precipitation datasets and the lack of streamflow observations for several regions [36].

With the rapid development of satellite remote sensing technologies, varieties of models and approaches have been conducted as ET mapping methods over the past decades. Satellite remote sensing offers a reasonable estimation of land surface variables, such as albedo, vegetation index (VI), and surface temperature (Ts), but does not measure ET directly [29]. The satellite remote sensing techniques that are widely used to retrieve ET for long-term variability at different scales include the surface energy balance model (SEB, i.e., one-source model and two-source model) [37,38], physical models (such as the Penman–Monteith model) [39,40], and empirical models [41]. In the surface energy balance model, ET is determined by the residual term of the surface energy balance. The land surface temperature (LST) is a crucial RS parameter in SEB models; however, there are some limitations to applying LST data to estimate ET on a global scale [42,43]. They have been widely used to estimate ET with routinely available meteorological data and relatively high resolution (i.e., Landsat and the advanced spaceborne thermal emission and reflection radiometer, or so-called ASTER) and modest resolution satellite data (such as the moderate resolution imaging spectroradiometer, or so-called MODIS). For example, recently, Numata et al. [25] explored forest ET dynamics under extreme drought in Southwestern Amazonia using the mapping evapotranspiration with high resolution and internalized calibration (METRIC) model. A large number of remote sensing datasets have emerged and provide a useful method to obtain spatiotemporal continuous ET information, while studies also indicate that a 15–30% relative error remains between these RS-based ET products [44–46]. Different from the SEB model, the physical model and empirical model are flexible to adopt satellite observations and meteorology reanalyzing data that could be applied to estimate ET at a global scale, showing generally reasonable results [29]. Recently, large artificial intelligence (AI) models have been applied in the field of ET estimation, such as the support vector machine (SVM), random forest regression (RF), the artificial neural network (ANN), and the wavelet random forest (WRF) model [47–49]. Compared with the conventional empirical model, the AI models show a superior performance in the selection and identification of input variables, as well as finding the optimal correlations between variables [50].

In addition, global ET can be estimated by land surface models (LSMs), for instance, as part of the Global Soil Wetness Project 2 dataset (GSWP-2) [51] and the Global Land Data Assimilation System dataset (GLDAS) [52] products. The reprocessed GLDAS Version 2 (GLDAS-2) data products have been released and could offer important references to support the response of ET to extreme drought impacts. The Monin Obukhov similarity theory (MOST) provides the fundamental equations to estimate ET for LSMs, calculated as the sum of the vegetation transpiration, vegetation evaporation, and soil evaporation [29]. Based on the observation-based forcing data and satellite RS data, LSMs can be a useful approach to derive long-term ET data with a relatively higher spatiotemporal resolution at regional and global scales [53,54]. Nevertheless, intercomparisons of LSMs showed

large discrepancies because LSM simulation is susceptible to the model parameters, model structure, and the reliability of the land surface forcing data.

With the increase in large satellite RS data and ground-based observations, the data assimilation method can be applied to improve the accuracy of LSMs by assimilating multisource data into the model. Data assimilation is a method for integrating a number of groups of data, such as ground-based observations, RS data and products, and surface mete-oritical forcing data, into model simulations in order to achieve the best model estimations. Over the last two decades, various global, national, and regional land data assimilation systems (LDASs) have been developed and have generated many useful surface flux products that have improved the ability of LSMs to predicted ET [55] (i.e., ERA-Interim [56], MERRA [57], and NCEP-DOE [58]). However, many challenges remain in the quality of input data, spatial mismatches between model grids, and ground-based observations, that may impact the accuracy of the ET simulation results [55].

*2.2. Comparisons of the Methods*

There is no agreement on which approach or product is the best because all of them have certain strengths and limitations and all have proved to be of acceptable quality for drought impact research in different situations. Ground-based observation measurements can obtain relatively accurate estimates of ET at the point scale, while their applicability may be limited over large areas [59]. The water balance approach is simple and theoretically reliable, allowing for basin areas of long-term accurate ET estimates with the limitation of a relatively low temporal resolution (yearly). Therefore, most models and products evaluate the performance of ET estimation by comparison with the flux site observations and water budget-derived ET. Satellite remote sensing techniques have been considered the most feasible way to provide spatial coverage, cost-effectiveness, and a reasonably accurate method for estimating ET at the regional to global scales. The SEB model is one of the earliest RS models to estimate ET, while the accuracy largely relies on the reliability and quality of the thermal remote sensing data. In addition, the SEB model is only reliable in clear-sky conditions because of the utilization of thermal RS data. Empirical models are easy to apply with the minimum climate inputs, but they usually lack a solid physical basis and depend significantly on calibration. LSMs and data assimilation methods provide a useful way to obtain long-term ET data at a regional to global scale and can be used to forecast ET dynamics in the future. Nevertheless, the quality of input data and model structure continue to have an impact on the accuracy of ET simulation.

Numerous studies have been conducted to compare the performance of different global ET products. Mueller et al. [46] compared the seasonal variations and multiyear mean spatial patterns of the LandFlux-EVAL datasets based on observational data, including satellite-based products, LSM simulations, and reanalysis products. The results revealed that the investigated datasets demonstrated similar spatial patterns despite the differing observation constraints, while the IPCC AR4 simulations displayed an underestimated ET within several continents. Jimenez et al. [60] supplemented Mueller's research, which focused on the latent and sensible flux, and the results demonstrated that RS-based products exhibit a higher correlation between the latent heat flux and the net radiation compared with the off-line LSM models and reanalysis products. Aiming to develop high-quality and consistent global ET datasets, Mueller et al. [61] derived monthly global ET benchmark synthesis products by merging 40 distinct datasets, including diagnostic datasets, LSMs products, and reanalysis products. The synthesis products also provide distinct statistics for muti-data, and the annual mean ET shows similar interannual variations in all categories, while the merge products based on diagnostics and re-analyses exhibit the largest interquartile ranges. Above all, to reduce the uncertainties in quantifying ET and to increase the accuracy of input data, a substantial effort should be made to establish improved multi-sensor RS measurements and observation networks, especially in areas with poor spatial coverage. An overview of current global ET datasets, including

observation-based products, satellite RS-based products, LSMs, and reanalysis products, is provided in Table 1.

**Table 1.** Global terrestrial ET from different data sources.

| | Name | ET Scheme/LSS Scheme | Input Data | Reference |
|---|---|---|---|---|
| Observation | AWB-ETH | Atmosphere water balance | GPCP, ERA-Int | Muller et al., 2011 [62] |
| Remote Sensing | MPI-BGC | Empirical | CRU, GPCC, AVHRR | Jung et al., 2009 [63] |
| | PRUNI | Pemnan–Montheith | ISLSCP II | Sheffield et al., 2010 [64] |
| | PT-JPL | Priestler–Taylor | AVHRR, ISLSP II | Fisher et al., 2008 [65] |
| | CSIRO | Pemnan–Montheith | GPCC | Zhang et al., 2010 [66] |
| | MODIS | Pemnan–Montheith | GMAO, MODIS | Mu et al., 2011 [43] |
| Reanalysis | ERA-Interim | TESSEL | ECMWF | Dee et al., 2011 [56] |
| | MERRA | GEOS-5 Catchment LSM | Observations from EOS | Rienecker et al., 2011 [57] |
| | NCEP_DOE | NOAH | NOAA, OAR, ESRL PSD | Kalnay et al., 1996 [58] |
| | JRA-25 | SiB | JMA | Onogi et al., 2007 [67] |
| LSMs | GSWP-2 | Aerodynamic, Penman–Monteith | ISLSCP II | Dirmeyer et al., 2006 [51] |
| | GLDAS | NLDAS | Observation, Reanalysis | Rodell et al., 2004 [52] |
| | WaterMIP | Aerodynamic, Penman–Monteith, Priestler–Taylor | WATCH forcing data | Haddeland et al., 2011 [68] |
| | MERRA-LAND | Penman–Monteith | MERRA reanalysis data | Reichle et al., 2011 [69] |
| | NOAN-PF | Penman–Monteith | NCEP-NCAR, TRMM, GPCP | Sheffield et al., 2006 [70] |

## 3. Evapotranspiration Feedbacks to Drought

While drought has a significant impact on ecosystems [71], the response of ET to extreme droughts remains uncertain, and results vary among studies, possibly due to different climates, durations of water deficits, and ET assessment methods [72,73]. Wang et al. [29] reviewed the different responses of ET to drought under various situations. In water-limited regions, soil moisture provides the first-order control on ET [20]. Extreme droughts would limit the availability of water for both soil and rainfall interception evaporation and plants' transpiration by reducing precipitation and soil moisture, especially in transitional regions [20]. Increasing air temperatures and vapor pressure deficits (VPD) enhance ET due to the linear relationship between the temperature and the mean kinetic energy of liquid molecules [21]. In addition, the effect of VPD is nonlinear because an increased VPD will decrease stomatal conductance to minimize water loss via closing stomata [24]. The issue is also supported by the finding that about 29% of the global domain shows a significant increase in ET from 1982 to 2013, mainly driven by the increased air temperature and VPD [73]. In the equatorial region, solar radiation is the dominate factor of ET [74,75]. Studies have reported that the Amazon rainforest sustains elevated ET during drought periods because of fewer clouds [76,77].

Terrestrial ET could rapidly decrease due to the widespread plant mortality during a severe drought, and the decline of terrestrial ET could last for years due to changes in the ecosystem structure and functionality [71]. Severe droughts can affect the ecosystem, which involves changes in the species and the community scales and alterations in soil properties. In the species scale, extreme drought result in various changes in species structure, physiology, phenology, and plant defense [71]. For instance, drought can alter the leaf morphology, root density, and growth rate and can alter plant species toward drought-tolerant species. Few studies have explicitly investigated the duration of drought impacts on ET. In general, compared to the C-cycle-related drought impacts, which last for several years, the flux parameters are back to their pre-drought levels within a year [78,79]. Additionally, Wu et al. [80] and Zhang et al. [81] reported that the duration of recovery to pre-drought level tends to be longer for woody species and forests compared to herbaceous/non-woody species and grasslands. Plant mortality may occur during and after the drought, lead-

ing to a rapid decline in ET [82]. Plant mortality is typically caused by a combination of several factors, for instance, diseases related to pests and pathogens, climate stress, and wildfires [83]. McDowell et al. [84] developed three hypotheses of plant mortality during droughts: (i) hydraulic failure, where large water losses result in xylem cavitation; (ii) carbon starvation, where stomatal closure prevents carbon uptake in order to reduce water losses; and (iii) biotic attack, in which changes in insects and pathogen populations in warmer temperatures cause amplified plant damage and mortality. Plant mortality has a long and lasting effect on the ecosystem and community scale. Furthermore, the feedback of ET may amplify the severity of the drought because (i) a higher ET due to atmospheric conditions drying may exacerbate soil moisture depletion [12]; (ii) a higher ET may also amplify runoff and storage anomalies during drought episodes [12]; and (iii) SM negative anomalies can exacerbate precipitation anomalies, which may act to amplify droughts [85].

The timing, severity (duration and intensity), and frequency of droughts may also have an impact on the results of ET's response to drought. In general, increasing drought severity (with longer duration and higher intensity) tends to hasten soil moisture depletion and plant mortality, resulting in a lager reduction in ET. The timing of the drought matters for the degree to which ET is affected by the drought. For example, D'Orangeville et al. [86] demonstrated that droughts in the early growing season have induced the greatest reductions in tree growth. In addition, Huang et al. [87] also indicated that extreme droughts that occur during the dry season have more lasting and severe effects compared to droughts that occur during the wet season. Furthermore, there are several studies suggesting that drought resistance is probably decreased by recurrent droughts [88,89], but there are also arguments that it is elevated [90].

The response of terrestrial ET to elevated $CO_2$ is expected to increase the uncertainty of the ET changes during droughts because elevated $CO_2$ tends to increase water use efficiency which may offset the effects of drought to some extent [91,92]. By regulating the structure and physiology of vegetation, increasing $CO_2$ concentrations exert two opposite impacts on ET [93]. On the one hand, elevated $CO_2$ decreases the stomatal conductance, resulting in less water being transpired per unit leaf area and potentially increasing water use efficiency [94–96]. On the other hand, several field experiments reported that the vegetation leaf area may increase contributing to a rising leaf biomass, resulting in higher ET that may partly offset the decreased stomatal conductance [95,97]. Both the increased and decreased responses of ET to elevated $CO_2$ have been reported for different species and climates [25–27]. However, the extent to which the effects of extreme drought on ET can be offset by elevated $CO_2$ fertilization still remains uncertain and debatable. Morgan et al. [98] found that elevated $CO_2$ reversed the negative effects of water constraints in semiarid grasslands using FACE experiments. However, Yuan et al. [99] found that rising VPD caused a large terrestrial GPP decline in the late 1990s which offset the impacts of the elevated $CO_2$ fertilization. The duration and intensity of droughts and ecosystem categories are considered to determine the impacts of elevated $CO_2$ on drought resistance [100].

## 4. Mechanisms of Terrestrial ET Variation

Terrestrial ET is controlled by complex biological and physical processes. Zhang et al. [73] divided the physical process into three components: demand, energy, and supply, and analyzed average values in these three controls over 32 years (from 1982 to 2013) using meteorological reanalysis datasets. The results showed that over 49% of the global domain is dominated by the water supply, whereas over 32% and 19% of the global domain was dominated by the available energy and atmospheric water demand, respectively [73]. In addition to these physical constraints, vegetation also exhibits a significant impact on ET. However, the relative importance of ET changes with different biological and physical processes is still unknown. Therefore, in this section, we review the control factors of ET changes during droughts in different situations.

### 4.1. Water Supply

Water supply, specifically, the soil moisture content (SMC), plays an important role in terrestrial ET by influencing both soil evaporation and plant transpiration during droughts. An important framework based on the separation of the soil moisture content describes the determination of ET under various climate regimes. Under wet conditions, corresponding to the SMC values above the critical value $\theta_{CRIT}$, ET is independent of the water supply and is limited by energy. When the SMC id below $\theta_{CRIT}$, the SMC is the dominant control of ET. Under water-stressed situations, when SMC decreases, the soil moisture potential becomes more negative, and ET decreases through limiting the diffusion of water molecules from roots. Under extremely dry situations, the SMC falls below the wilting point $\theta_{WILT}$ ($\theta_{WILT} \leq \theta \leq \theta_{CRIT}$), and ET drops to near zero and no longer changes. According to this useful framework, three climate regimes can be defined based on the ET changes associated with the soil moisture content: dry ($\theta < \theta_{WILT}$) and wet ($\theta > \theta_{WILT} < \theta_{CRIT}$) climate regimes, in which the soil moisture content is independent of ET, and a transitional climate regime ($\theta_{WILT} \leq \theta \leq \theta_{CRIT}$), in which the soil moisture content strongly constrains ET variability.

This concept has been widely used in LSMs as a "bucket model", in which the general expression is as follows:

$$ET = \beta ET_{POT} \tag{1}$$

$$\beta = \frac{\theta - \theta_{WILT}}{\theta_{CRIT} - \theta_{WILT}} \quad \text{for} \quad \theta_{WILT} \leq \theta \leq \theta_{CRIT} \tag{2}$$

where $\theta$ is the soil moisture content, $\beta$ is the evaporation factor or stress factor, $\theta_{CRIT}$ is the critical value of the SMC which determines whether the SMC is the first-order control for ET, and $ET_{POT}$ is the potential evaporation.

This formula indicates that the soil moisture content is a key constraint for ET changes in transitional regimes, for instance, grasslands and shrublands, where the soil water content is always limited. Both observations and satellites confirm the control of ET by the soil moisture content in the dry–wet transition regime. For example, Teuling et al. [101] found that water-limited sites show a decrease in the soil moisture content and subsequent evapotranspiration and an almost linear dependency of ET on the soil moisture content. Due to the lack of soil moisture observations at all depths across large areas, precipitation is frequently used instead of the soil moisture content. Notably, underground water and irrigation also have a nonnegligible impact on the available water stress, especially in semiarid and arid areas [29].

### 4.2. Atmospheric Evaporative Demand

The atmospheric evaporative demand represents the demand for water from the atmosphere (i.e., its evaporating or drying power) [102]. The potential for atmospheric water evaporation can be divided into two parts [103]: (i) radiative, representing the available energy to evaporate water, which is primarily determined by solar radiation; and (ii) aerodynamic, representing the ability of the atmosphere to carry water, which is primarily determined by climate parameters, including air temperature, humidity, and wind speed.

During droughts, rising air temperatures can enhance the transpiration rate by increasing the VPD, which is always considered as a metric of atmospheric evaporative demand [24,103,104]. However, the impacts of rising temperatures and vapor pressure deficits on terrestrial ET are nonlinear because vegetation tends to close its stomata to prevent excessive water loss when the atmospheric demand is high [24]. Katul et al. [105] reported that the transpiration rate increases with the vapor pressure deficit, increasing to a critical maximum value, after which it remains stable or even declines as a result of stomatal closure.

Penman [103] developed the widespread combination equation for potential ET, which is generally expressed as:

$$\lambda E_p = \frac{\Delta(R_n - G) + \rho_a C_\rho D / r_a}{\Delta + \gamma},\tag{3}$$

where $\rho_a$ is the air density, $\Delta$ is the slope of the saturated vapor pressure curve at a given temperature, $C_\rho$ is the specific heat of the air constant pressure, and $\lambda$ is the psychrometric constant. The variables $\Delta$ and $\gamma$ are functions of temperature, $R_n$ is the net radiation, $G$ is the soil heat flux, and lastly, $r_a$ is the aerodynamic resistance of the heat flux.

The Penman algorithm (Equation (3)) is a physis-based equation for estimating ET under homogeneous and unrestricted water supply conditions, so-called potential evaporation (PET). The PET predominantly determines terrestrial ET through the climate factors listed in the formula, which include solar radiation, humidity, wind speed, and temperature. The relationship between the ET variability and climate changes or climate variability explains why the earth's dimming or brightening, global warming, and calming of winds may all have an impact on terrestrial ET.

### 4.3. Energy

The process of ET is controlled by the energy exchange at the plant's surface and is constrained by the amount of energy available. Terrestrial ET is a transformation of water from the liquid to the vapor phase, which requires substantial energy. The driving force of moving water vapor molecules away from the evaporating surface is the water vapor pressure difference between the surrounding atmosphere and the evaporating surface [106]. Solar radiation and, to a lesser degree, the ambient temperature provide this energy [107].

The latent heat flux ($\lambda ET$), which represents the evapotranspiration fraction, can be derived using the energy conservation principle when all other variables are known. All flux energies can be described using the terms of the energy balance equation that can be written as [107]:

$$\lambda ET = R_n - G - H,\tag{4}$$

Numerous studies have shown that energy is the controlling factor in high-latitude ecosystems and tropical ecosystems [73]. High-latitude ecosystems are characterized by slow transpiration due to the lower solar radiation and temperature [108]. Droughts alleviate the critical climatic constraints to ET at high latitudes, such as increased solar radiation, temperature, and VPD [109,110]. Humid regions are characterized by a sufficient water supply and a shallow water table that can be reached by forests' deep roots during droughts [111]. Solar radiation is the predominant control of ET variability during droughts in Amazon rainforests [74,75]. During droughts, less cloud cover permits more solar radiation to reach the land surface, resulting in higher ET rates [112]. These observations and turbulent flux measurements support this hypothesis [74–76,113].

### 4.4. Physiological Limitations of Vegetation

Aside from the physical constraints, terrestrial ET is also constrained by vegetation, which connects soil water to the atmosphere via leaf stomata and roots. Numerous studies suggest that the interannual variation in vegetation activity primarily controls interannual changes in ET during the growing season [114,115], whereas others confirm vegetation's influence on the trends and the spatiotemporal patterns of ET at local to global scales [73,116–118]. Therefore, it is crucial to investigate how plants' physiological activities control the changes in ET during droughts.

Plants exhibit a variety of defense mechanisms to deal with drought stress. The major mechanisms of drought resistance include restricting water losses by raising the diffusion resistance, increasing water uptake from the soil with dense and deep root systems, and adjusting osmotic processes in tissues [16,119]. Plants have devised several strategies to deal with drought stress: (i) drought escape (shorten the life cycle or growing season

by accelerating the phenological process, allowing plants to escape form drought stress), (ii) drought avoidance (endurance with high water potential in tissues and prevention of organ damage), and (iii) drought tolerance (endurance with low water potential in tissues, while sustaining vegetation growth during droughts) [16,120]. The drought avoidance strategy maintains a high internal water content by reducing water loss from plants by decreasing transpiration via stomatal closure and reducing leaf area as well as increasing water uptake from the soil through a prolific and extensive root systems [121,122]. All these plant feedback processes have trade-offs between the risk of water depletion and carbon cost [123]. Different types of plants have different mechanisms for coping with drought, and they may combine several coping mechanisms to survive.

The most important modification to account for the physiological controls of terrestrial ET is the Penman–Monteith (P-M) equation [124], introduced by Monteith under the "big-leaf" assumption. This equation explains the movement of water away from the collectively saturated surfaces of plant leaves to the air through canopy-scale resistance. It is stated as:

$$\lambda ET = \frac{\Delta(R_n - G) + \rho_a C_p D / r_a}{\Delta + \gamma(1 + \frac{r_s}{r_a})},$$ (5)

where $r_s$ is the stomatal resistance. The Jarvis–Stewart equation was the first model to determine the stomatal resistance ($r_s$), combining biological and environmental controls in a multiplicative way, which can be expressed as [125–127]:

$$r_s = g_{smax} f(PAR) f(T) f(VPD) f(\psi) f(CO_2),$$ (6)

where $T$ is the temperature; $PAR$ is the photosynthetically active radiation; $\psi$ is the leaf water potential, which directly relates soil water content, $g_s = 1/r_s$; and $g_{smax}$ is the maximum conductance without factor restrictions. These equations (Equations (5)–(6)) indicate that stomatal resistance ($r_s$) is a key constrain for ET variability and strongly impacts by the climate variability, including sunlight, temperature, leaf water potential, $VPD$, and $CO_2$ concentration.

In summary, drought's impacts on terrestrial ET depend on the duration and degree of the reduction in the soil moisture content and precipitation gradient as well as the original climate conditions and plant species. The impact of physics and biology on terrestrial ET is not independent. The physical constraints directly affect vegetation growth and modify the control of vegetation on ET. Plants' physiological activities control ET through stomatal closure and root systems and alter the physical control through blocking precipitation and light and creating turbulence. However, how biological and physical mechanisms interact, the relative importance of these processes, and how these components will change remains uncertain in the changing world.

## 5. Conclusions and Outlook

Extreme drought events have had a strong impact on terrestrial ET during recent decades. Changes in ET strongly affect temperature, precipitation, soil moisture, and other hydrometeorological factors through land–atmosphere interactions, ultimately altering the spatiotemporal variations in water source availability and even regional and global climate systems.

We reviewed the current understanding of the terrestrial ET response to drought events and investigated the mechanisms of terrestrial ET variations under drought stresses. In conclusion, the response of terrestrial ET to drought depends on the timing and severity (duration and intensity) of the drought, the original climate conditions, as well as plant species. Terrestrial ET is controlled by complex physical and biological processes. The physical process of controlling ET can be separated into three parts: (i) water supply (as a function of soil moisture and precipitation); (ii) atmospheric evaporative demand (as a function of humidity, wind speed, and air temperature); and (iii) energy (as a function of solar radiation). The biological process control of ET is mainly exerted by vegetation

activities (as a function of leaf area index, leaf area density, leaf water potential, rooting depth, $CO_2$, and humidity). However, large uncertainties remain in estimating the ET response to drought deriving from large discrepancies from different ET products and simulation methods. To accurately estimate the response of ET to drought events, we need more accurate simulations of ET dynamics through the use of reliable ET products combined with observation verification.

Overall, this review provides a comprehensive investigation of the terrestrial ET response to extreme drought and the underlying mechanism of terrestrial ET changes during droughts. This work will significantly improve our understanding of the earth system and aid in the development of more effective water resource management strategies under climate change.

**Author Contributions:** Writing—Original draft preparation, Q.-L.H. and W.-Y.S., writing—review and editing, J.-L.X. and W.-Y.S. All authors have read and agreed to the published version of the manuscript.

**Funding:** This research was funded by the National Natural Science Foundation of China (No. 41975114), the Chongqing Outstanding Youth Science Foundation (No. cstc2021jcyj-jqX0025), JSPS BRIDGE Fellowship (No. BR221301), and the Chongqing elite-innovation and entrepreneurship demonstration team (to Weiyu Shi).

**Institutional Review Board Statement:** Not applicable.

**Informed Consent Statement:** Not applicable.

**Data Availability Statement:** No new data were created or analyzed in this study. Data sharing is not applicable to this article.

**Conflicts of Interest:** The authors declare no conflict of interest.

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
