# Peer review of "Responses of Terrestrial Evapotranspiration to Extreme Drought: A Review"

_water, doi:10.3390/w14233847_

Round 1
Reviewer 1 Report
This article comprehensively overviews the current findings about the response of terrestrial ET to extreme drought and the mechanisms of terrestrial ET variation. The concluded the current difficulties and solutions in this field, which has significant implications for water source management. This is a nice work and falls in the scope of this journal. The manuscript is in good organization and writing. However, after reviewing, there are still some concerns that need to be addressed before acceptance.
In section 2, make the description more clearly for the aims of this section. Changing the title of the section to be more suitable for the content.
I section 2.2, add explanations to the relationship between multiple ET products and simulation methods and the response of terrestrial ET to extreme droughts.
Some sentences or phases should be improved, and they are listed below.
(1) Line 14-22: the results of this review is not clear enough, what are the points? It looks scattered now.
(2) Line 14-16: “The response of terrestrial ET to extreme drought is affected by various factors including the duration and intensity of drought, original climate conditions, as well as plant species and their developmental stages.”
the effect of vegetation developmental stages on the response of ET to extreme drought was not mentioned in this article.
(3) Line 187: the “soil type” is not mentioned in the following article.
(4) Line 194: Here the word "however" is repeated with the previous sentence.
(5) Line 200-204: the sentences represent “the implication of ET feedback to extreme droughts” seems does not relevant with this paragraph content, it would be better to move to the second paragraph of Part 3.
(6) Line 257-259: “However, the relative effects of ET by different biological and physical processes vary on spatial and temporal scales.”
this sentence seems out of place. It should write “what is the biophysical process” first, and then point out “the biophysical process could change with time and space”.
(7) Line 246-247: Add several references would be better here.
(8) Line 292-294: The sentence could be more clearly expressed here. The statement is confusion “decrease in soil moisture drying”.
(9) Line 337-339: the content is not in accordance with the content of this section.
(10) Line 415-422: It seems a consensus conclusion from the past literature, the results need to be rewritten summarized from this study, same as the Abstract section.
Reviewer 2 Report
The paper requires major modifications before it is processed:
-Motivation and innovation of the present research towards review of responses of terrestrial evapotranspiration to extreme drought are missing!!!
-Final paragraph of introduction section is generally related to the what will happen in the paper. This section is missing.
-Why did not authors review application of artificial intelligence models and remote sensing techniques into prediction of evapotranspiration???
-Why did not authors compare the performance of various techniques from literatures in terms of statistical measures (like R and RMSE)???
-Authors need to select the best model to forecast ET.
-All the models in this paper need to describe major merits and drwabacks in the new sections of paper.
Reviewer 3 Report
Responses of Terrestrial Evapotranspiration to Extreme Drought:A Review
Dear authors,
Thank you for submitting your manuscript to Water. First, I checked the similarity report using Turnitin and found that the scores are 27% which is high for a journal paper. You should be able to keep that under 20%. In addition, I have concerns over a particular paper which you have 4% similarity.
“Zhenzhong Zeng, Liqing Peng, Shilong Piao. "Response of terrestrial evapotranspiration to Earth's greening" , Current Opinion in Environmental Sustainability, 2018”
You have to seriously think about this issue and arrange your manuscript accordingly. A journal accepts 2% from a single paper. If this not rectified in your revisions, your manuscript can be rejected. I have attached the Turnitin report for your reference.
Authors have done some extensive reviews; however, they have failed to showcase the importance of this review. Why someone has to read your review paper if it is published? This is highly missing in this review.
Abstract is quite acceptable. Could have showcased the importance of this review more.
Introduction –
“Therefore, a slight change in ET caused by extreme droughts will have a significant impact on the regional water resources and climate simulation via land-atmosphere interaction.” - This statement has to be either supported by numerical findings or by a solid reference.
“This review summarizes the current understanding of ET response to drought stress and highlights the control factors of ET changes to extreme droughts under different conditions.” – What is the research gap to do this review?
I do not understand why you need such a definition? Any reason to title the subtopic like this? This is well known among research community. “2.1. Drought Definition”
Line 181 – Probably in written in Chinese?
Authors should finalize the review stating the importance of reading this paper.

Round 2
Reviewer 2 Report
Accept as is
Reviewer 3 Report
Revisions are accepted.